# Technical Efficiency Evaluation of Primary Health Care Institutions in Shenzhen, China, and Its Policy Implications under the COVID-19 Pandemic

**DOI:** 10.3390/ijerph20054453

**Published:** 2023-03-02

**Authors:** Shujuan Chen, Yue Li, Yi Zheng, Binglun Wu, Ronita Bardhan, Liqun Wu

**Affiliations:** 1Shenzhen Health Development Research and Data Management Center, Shenzhen 518028, China; 2Department of Architecture, University of Cambridge, Cambridge CB2 1PX, UK; 3Department of Structural Reform and Primary Health Care, Shenzhen Municipal Health Commission, Shenzhen 518031, China

**Keywords:** primary health care, COVID-19, data envelopment analysis, Malmquist index model, Tobit regression

## Abstract

(1) Background: Primary health care institutions (PHCI) play an important role in reducing health inequities and achieving universal health coverage. However, despite the increasing inputs of healthcare resources in China, the proportion of patient visits in PHCI keeps declining. In 2020, the advent of the COVID-19 pandemic further exerted a severe stress on the operation of PHCI due to administrative orders. This study aims to evaluate the efficiency change in PHCI and provide policy recommendations for the transformation of PHCI in the post-pandemic era. (2) Methods: Data envelope analysis (DEA) and the Malmquist index model were applied to estimate the technical efficiency of PHCI in Shenzhen, China, from 2016 to 2020. The Tobit regression model was then used to analyze the influencing factors of efficiency of PHCI. (3) Results: The results of our analysis reflect considerable low levels of technical efficiency, pure technical efficiency, and scale efficiency of PHCI in Shenzhen, China, in 2017 and 2020. Compared to years before the epidemic, the productivity of PHCI decreased by 24.6% in 2020, which reached the nadir, during the COVID-19 pandemic along with the considerable reduction of technological efficiency, despite the significant inputs of health personnel and volume of health services. The growth of technical efficiency of PHCI is significantly affected by the revenue from operation, percentage of doctors and nurses in health technicians, ratio of doctors and nurses, service population, proportion of children in the service population, and numbers of PHCI within one kilometer. (4) Conclusion: The technical efficiency significantly declines along with the COVID-19 outbreak in Shenzhen, China, with the deterioration of underlying technical efficiency change and technological efficiency change, regardless of the immense inputs of health resources. Transformation of PHCI such as adopting tele-health technologies to maximize primary care delivery is needed to optimize utilization of health resource inputs. This study brings insights to improve the performances of PHCI in China in response to the current epidemiologic transition and future epidemic outbreaks more effectively, and to promote the national strategy of Healthy China 2030.

## 1. Introduction

The Chinese government has committed to achieving the plan of “Healthy China 2030” with promoting universal health coverage as a major ambition in 2016 [1]. As part of China’s health-care endeavor to increase health equity and coverage, primary health-care institutions (PHCI) play an important role as the “gatekeeper” of Chinese residents’ health level. PHCI in China were designated to undertake the responsibility of delivering general medical services and basic public health services to residents at community level. These services regularly include managing chronic diseases, reception of outpatients and emergency visits, providing home visits and educating services, children vaccinations, public health issues reporting and physical examinations, and establishing resident health records [2]. Over the past 10 years, the health-reform launched by the Chinese government has made substantial efforts in improving the PHCI, as the foundations of the integrated health delivery system in China, to provide citizens with equitable routine health services [3,4]. What is more, the recent coronavirus disease 2019 (COVID-19) outbreak underlined the essential role that PHCI could play in screening and monitoring the epidemic.

In the face of COVID-19, PHCI were designated responsible for control and prevention of COVID-19 under the Chinese governments’ strict non-pharmacological interventions characterized by strict community management. They were allocated duties for conducting COVID-19 screening and referral, monitoring, education, and publicity by the National Health Commission [5], which alleviated the pressure faced by general hospitals and the whole health system. The fever patients or suspected COVID patients were transferred by PHCI to the specialist fever clinics at the higher-level hospitals to receive treatment, based on the government regulations, which promoted the effective control of the pandemic. Over 90% of PHCI in China have carried out these new tasks and made positive impacts on monitoring suspected COVID-19 cases and controlling the epidemic at community level [6]. However, these achievements could not be made without the considerable investments and allocations of health professionals and financial resources. Although the application of PHCI in the control of the unprecedented pandemic is relatively effective, some issues have also been exposed.

The suboptimal efficiency of PHCI was brought to the fore by the COVID-19 epidemic, as a challenging test for the development of the integrative health systems advocated by China in recent years. Due to substantial additional workload as part of the epidemic control, PHCI experienced shut-down from their routine care on other health conditions. The access to the routine functions of PHCI such as outpatient visits and emergency visits were significantly affected. In addition, despite the health resources inputs have increased in the past years especially during the pandemic, large quantities of PHCI have been still confronting issues such as deficient manpower and skills, insufficient medical supplies, flaws in public information systems, and decreased patient visits [7]. They are basically not producing efficient and economical outputs compared to their corresponding substantial inputs of resources. PHCI were facing the challenge of efficiently handling the core responsibilities in managing emerging infectious diseases such as COVID-19 as well as maintaining the routine care on other health conditions such as hypertension simultaneously. What is more, as there is the approach of an aging society in China by 2030, national burdens from chronic disease continues its relentless increase, which further calls for effective PHCI. To improve the performances of PHCI as a decisive bedrock of the integrated health system during events such as an epidemic and aging era, it is essential to escalate the service efficiency of PHCI.

In the past decades, the Chinese government has made considerable efforts to advance PHCI. For example, the funding allocated to PHCI has increased by more than ten times, from 19 billion in 2008 to 197 billion in 2018, by the Chinese government [8]. The Healthy China 2030 Plan announced in 2016 proposes the equalization of basic public health services by emphasizing prevention and primary care, which further bolsters the role of PHCl [9]. In May 2021, the acceleration of the improvement of the basic infrastructure of community health centers and township hospitals was issued by the State Council, as one of the key tasks to deepening the reform of the health system [10]. In November 2022, the Chinese government issued new guidelines easing some of its strict COVID policies, including relaxation of testing requirements, travel restrictions, and quarantine for COVID patients [11]. PHCI in a city such as Shenzhen shoulder the new tasks of diagnosis and treatment of suspected COVID-19 patients by setting up over 2000 fever clinics within the communities [12]. The fever patients no longer need to be transferred to the fever clinics at the higher-level hospitals to receive treatment, which further mitigates the stress of residents and hospitals and enhances the integrated health system. We can see that PHCI play a more and more imperative role in tackling the responsibility of routine basic health care as well as controlling the epidemic in the post-epidemic era. However, these achievements were achieved with the insurmountable inputs of medical and human resources, which, on the other hand, might lead to the waste of excessive resources due to deficient service efficiency. It is crucial to build strong PHCI’s optimal service efficiency to construct an effective integrated health delivery system in China.

However, the status of PHCI’s technical efficiency and the underlying influencing factors in the recent years are unclear. Some Chinese scholars evaluated the performances of PHCI during the pandemic using only health resources inputs and service capacities without considering the outputs and subsequent efficiency [13,14]. Studies assessing the impacts of COVID-19 on PHCI focused on the physical and mental health of PHCI practitioners [15,16,17], and the roles and services of PHCI in response to the pandemic [14,18,19]. Most of them adopt qualitative methods rather than quantitative methods, and they merely measure changes in health resources of PHCI by several indicators, without considering broad health resource inputs, outputs, and underlying influential factors with service efficiency of PHCI. Efficiency studies are important for informed decision making to improve the performance of PHCI and reduce wastage, which invariably results in better allocation of health resources. A comparative table of the literature review from both aspects of theory and application is included to highlight the characteristics of current research (see Appendix A, Table A2). Thus, based on the analysis of previous studies, the research gaps comprise mainly three aspects: (1) the status of PHCI’s technical efficiency and the underlying influencing factors in China in recent years (before and after the pandemic) are still unclear; (2) previous research about the impact of COVID-19 on PHCI did not consider general health resource inputs, their relationship with outputs and underlying influencing factors of service efficiency of PHCI; (3) there is no study to date that has evaluated the efficiency changes of PHCI over a period that covers the outbreak of the COVID-19 pandemic to assess the impact of the COVID-19 pandemic using quantitative approaches. Therefore, this paper comprehensively analyzes the service efficiency of PHCI and identifies the influencing factors of improving the performance of PHCI before and after the epidemic.

There are several methods to estimate the efficiency of PHCI. One of the methods is data envelope analysis (DEA) which is a non-parametric approach first proposed by Charnes, Cooper, and Rhodes (CCR) for measuring technical efficiency (TE) of a decision-making unit (DMU) [20]. Compared to stochastic frontier analysis (SFA), DEA does not assume a functional form between the inputs and outputs and can handle multiple inputs and outputs [21]. The method of DEA has been widely used in production frontier analysis of factories, hospitals, and schools [22,23]. The analysis of DEA is commonly followed by a Malmquist index (MI) to measure the change of Total Factor Productivity (TFP) to indicate changes in institutional service efficiency over different time periods [24,25]. In addition, to study the changes in the technical efficiency of PHCI, this article also studied the key factors influencing the changes in technical efficiency of PHCI. Tobit regression is usually attached to DEA to identify the influencing factors that may contribute to the variation of technical efficiency between DMUs, and it is applied to analyze the influencing factors of the technical efficiency of PHCI before and after the pandemic.

Therefore, to study the technical efficiency of PHCI before and during the COVID-19 epidemic, we used DEA and MI to analyze the technical efficiency of PHCI in Shenzhen, China, from 2016 to 2020, treating each individual PHCI as a decision-making unit. Then, the Tobit model was used to estimate the key factors that influence the efficiency of PHCI. Since Shenzhen’s health policies were in line with those of China’s policies, the result of this study could well reflect and generalize the situation of PHCI in cities with middle to high level of economic development. We believe that the study of the technical efficiency of PHCI under the epidemic is informative and insightful for the Chinese government, in terms of developing superior policies for the advance of health resource allocation and service efficiency of PHCI in the face of a future unknown outbreak. It is also beneficial to the primary health practitioners to enhance the vulnerable loopholes in the process of general health services. What is more, this study puts forward practical insights for other regions and the government to better develop primary health care systems which can handle the core responsibility of managing the routine care of general health and emerging infectious diseases.

## 2. Materials and Methods

### 2.1. The Data Envelope Analysis

DEA is widely applied in the estimation of the technical efficiency of a set of DMUs because it is flexible when dealing with multiple inputs and outputs [26]. The efficiency score is defined as a ratio of the weighted sum of the outputs to the weighted sum of the inputs [27], which is derived from a non-parametric linear programming technique that recognizes an efficiency frontier on which only the efficient DMUs are selected [28]. A DMU is deemed to be technically efficient if it can produce maximum output from a given set of inputs.

Regarding the estimation of the efficiency frontier, CCR assumed production as constant returns to scale (CRS) which means any degree of increase in inputs will proportionately increase the level of output [20]. While Banker, Charnes, and Cooper (BCC) proposed another model which assumed the production as variable returns to scale (VRS), meaning any increase in the level of input will either increase or decrease the level of output [29]. In this way, the efficiency yield from the VRS model is the pure technical efficiency (PTE) that eliminated the scale effect. Scale efficiency (SE) is defined as the ratio of technical efficiency estimated based on the CRS model and the technical efficiency yield from the VRS model [29]. SE is represented as a measure of the extent to which a DMU deviates from an optimal scale, which should not be greater than one.

This study applied an input-oriented DEA model due to its concentration on minimizing the inputs for providing the given quantity of outputs. This model is suitable for the situations of health systems in China, because government and hospital can control the allocation of personnel and monetary inputs, rather than outputs such as the number of outpatient visits. Thus, an input-oriented DEA model produces better interpretations for policy making [30]. Other potential approaches such as Network DEA and Dynamic DEA were also developed to evaluate more complex production processes in different industries [31,32]. However, this study chooses the classic DEA approach because it suits the research questions better. Unlike studies that used Network DEA to evaluate efficiencies in different stages of production [32], this study does not consider multiple stages but focuses on the overall efficiency of PHCI since PHCI have simple and clear production processes. In addition, Dynamic DEA was not applied in this study, since the inputs and outputs of PHCI have clear temporal definitions and the production does not involve carry-over activities, efficiencies were evaluated independently each year. Later studies have also developed the bootstrap method to account for the influence of environmental and random factors on TE [33,34]. As is stated in these studies, the bootstrap method is preferred only when the number of DMU is limited. In this study, the number of DMU is large enough so that the bootstrap method is not considered necessary [33,34]. Therefore, the one-stage input-oriented DEA approach was adopted in this study based on the fact that it suits the research questions well.

The input-oriented CRS model to estimate technical efficiency is specified in Equation (Equation 1), and input-oriented VRS model to estimate PTE is specified in Equation (Equation 2).
(1)$$\begin{array}{ccc} \max & \;\frac{\sum_{r=1}^{q}\mu_{r}y_{rk}}{\sum_{i=1}^{m}\nu_{i}x_{ik}} & \begin{array}{c} s.t.\left\{\begin{array}{cc} \frac{\sum_{r=1}^{q}\mu_{r}y_{rj}}{\sum_{i=1}^{m}\nu_{i}x_{ij}}\leq 1 & \left(j=1,2,\cdots ,n\right)\\ \nu_{i}\geq 0 & \left(i=1,2,\cdots ,m\right)\\ \mu_{r}\geq 0 & \left(r=1,2,\cdots ,q\right) \end{array}\right. \end{array} \end{array}$$
(2)$$\begin{array}{ccc} \max & \;\sum_{r=1}^{s}\mu_{r}y_{rk}-\mu_{0} & \begin{array}{c} s.t.\left\{\begin{array}{cc} \sum_{r=1}^{q}\mu_{r}y_{rj}-\sum_{i=1}^{m}\nu_{i}x_{ij}-\mu_{0}\leq 0 & \left(j=1,2,\cdots ,n\right)\\ \sum_{i=1}^{m}\nu_{i}x_{ik}=1\\ \nu_{i}\geq 0 & \left(i=1,2,\cdots ,m\right)\\ \mu_{r}\geq 0 & \left(r=1,2,\cdots ,q\right) \end{array}\right. \end{array} \end{array}$$

Here, the subscript *k* indicates the DMU whose TE or PTE is being estimated. Then, $x_{ik}$ is the *i*th input of this DMU and $y_{rk}$ is the *r*th output of this DMU. For a given *k*, $x_{ij}$ is the *i*th input of the *j*th DMU, $y_{rj}$ is the *r*th output of the *j*th DMU. $\nu_{i}$ is the weight of the *i*th input, $\mu_{r}$ is the weight of the *r*th output. $\mu_{0}$ is a free variable ranging in (−∞,+∞) [35].

### 2.2. Malmquist Index

DEA estimates technical efficiency for a given year but is not applicable for comparing service efficiencies between different years or dealing with panel data. When technology evolution is the major facilitator of the improvement of the productivity, only measuring technical efficiency is insufficient. This factor is taken into account by the Malmquist index (MI) first proposed by Fare [36] who also used it as an efficiency index in the field of productivity analysis. Thus, the MI is applied to measure the change in TFP of DMUs over time in this study. In the model, an input-oriented distance function is defined as the difference between the actual outputs and the efficient outputs given the same set of inputs, i.e., the distance of a given DMU to the efficient production frontier [36]. MI is consequently the ratio of the distances using the set of inputs and outputs in year *t* and year t−1 compared to the same efficient production frontier. This efficient production frontier could be either that of year *t* or year t−1, so MI could be expressed as in either Formula (3) or (4). R. Färe et al. (1992) defined the distance function form of MI using the geometric mean of the two MI to calculate the productivity change (shown as Formulas (5) and (6)) based on the method proposed by Caves [37].

TFP could be decomposed into technical efficiency change (EC) and technological change (TC) through a mathematical transformation as shown in Formulas (7) and (8) [37]. Technical efficiency change can be further decomposed into pure technical efficiency change (PEC), which denotes the institutional management level, and scale efficiency change (SEC) as Formula (9) [38]. A TFP value > 1 indicates that productivity has increased in year *t* compared to year t−1. PEC value > 1 means that the management level has improved. If TC value > 1, the PHCI’s technology has improved. If SEC > 1, it signifies that, as the increase of the input factor, the production efficiency of PHCI has been escalated, and economies of scale have been achieved. Contrarily, if these indexes are smaller than 1, the corresponding efficiency is declining.
(3)$$\begin{array}{cc} MI_{\left(t-1,t\right)}^{t-1} & =\frac{D_{t-1}\left(x_{t},y_{t}\right)}{D_{t-1}\left(x_{t-1},y_{t-1}\right)} \end{array}$$
(4)$$\begin{array}{cc} MI_{\left(t-1,t\right)}^{t} & =\frac{D_{t}\left(x_{t},y_{t}\right)}{D_{t}\left(x_{t-1},y_{t-1}\right)} \end{array}$$
(5)$$\begin{array}{cc} MI(t-1,t) & =\sqrt{MI_{\left(t-1,t\right)}^{t-1}\times MI_{\left(t-1,t\right)}^{t}} \end{array}$$
(6)$$\begin{array}{cc} \; & =\sqrt{\frac{D_{t-1}\left(x_{t},y_{t}\right)}{D_{t-1}\left(x_{t-1},y_{t-1}\right)}\times \frac{D_{t}\left(x_{t},y_{t}\right)}{D_{t}\left(x_{t-1},y_{t-1}\right)}} \end{array}$$
(7)$$\begin{array}{cc} \; & =\frac{D_{t}\left(x_{t},y_{t}\right)}{D_{t-1}\left(x_{t-1},y_{t-1}\right)}\times \sqrt{\frac{D_{t-1}\left(x_{t-1},y_{t-1}\right)}{D_{t}\left(x_{t-1},y_{t-1}\right)}\times \frac{D_{t-1}\left(x_{t},y_{t}\right)}{D_{t}\left(x_{t},y_{t}\right)}} \end{array}$$
(8)$$\begin{array}{cc} \; & =EC(t-1,t)\times TC(t-1,t) \end{array}$$
(9)$$\begin{array}{cc} \; & =PEC(t-1,t)\times SEC(t-1,t)\times TC(t-1,t) \end{array}$$

### 2.3. Tobit Regression

Tobit regression was applied to analyze the influencing factors of PHCI’s TE during 2016 to 2020. Due to the fact that values of technical efficiency are censored at 1 and 0, using traditional regression models with the ordinary least square method is biased. Thus, this study followed the method of using the Tobit regression model adopted in previous technical efficiency analysis, which was first introduced by Tobin [39]. In addition, panel data were utilized in this model; thus, the mathematical formula for the Tobit regression model is written in Formula (10) as below:(10)$$\begin{array}{c} {y_{it}}^{*}={x_{it}}^{{}^{\prime }}\beta +\epsilon_{it}={x_{it}}^{{}^{\prime }}\beta +\mu_{i}+\nu_{it}\\ \begin{array}{c} y_{it}=\left\{\begin{array}{cc} 0 & {y_{it}}^{*}\leq 0\\ {y_{it}}^{*} & 0<{y_{it}}^{*}<1\\ 1 & {y_{it}}^{*}\geq 1 \end{array}\right. \end{array} \end{array}$$

Here, the subscript *i* indicates each unique PHCI across the five years, subscript *t* indicates the time period, $y_{it}$ is the value of TE for the *i*th PHCI in year *t*, ${y_{it}}^{*}$ is the latent unobserved TE value had the value not been censored theoretically, $x_{it}$ is the vector of explanatory variables, β is the vector of coefficients of the explanatory variables which is to be estimated, $\epsilon_{it}$ is the disturbance term which could be further divided into $\mu_{i}$ (Mu), a time-invariant PHCI-specific effect, and $\nu_{it}$ (Nu) the remaining disturbance [39]. In the results of Tobit regression, Ln(SigmaMu) and Ln(SigmaNu) which stand for the natural logarithm of the standard deviation of Mu and Nu, respectively, are presented. These two terms are estimated simultaneously with the rest of the coefficients in Tobit regression. The dependent variable is censored at 0 and 1 in this model.

### 2.4. Data and Variable Selection

For the DEA and the Malmquist index, data were extracted from the Shenzhen Health Statistical Information Report System during the period 2016–2020, and the Shenzhen Health Statistics Yearbook. The selection of input and output variables were based on the attributes of Chinese data and the requirements of the DEA and MI.

For input variables, a general consideration is to include material, personnel, and financial resources. However, previous studies argued that it is improper to include monetary variables such as expenditure to the input since it leads to the confusion of technical efficiency and allocation efficiency [34,40,41]. The inputs in this study are the number of licensed (assistant) doctors, the number of registered nurses, the number of other health technicians, and the number of equipment valued at more than RMB 10,000. Number of beds was not considered as an input because PHCI in Shenzhen are not designed to handle inpatient visits. Number of health technicians was further divided into doctors or assistant doctors, nurses, and other health technicians. An advantage of separating personnel based on their categories is to allow adjustment of weights while calculating the efficiency of each PHCI. Merging all categories of personnel added an improper hypothesis that the functions of different categories of personnel are identical. This is especially inappropriate for PHCI compared to hospitals because each PHCI may have goals of emphasis and would thus be disadvantaged in the DEA if PHCI were not allowed to adjust their weights of inputs.

Output variables were selected based on the functions and orientations of PHCI in China, as defined in the Guidelines for Service Capacity Evaluation of PHCI according to the National Health commission [42]. Meanwhile, another principle is that the outputs need to be comprehensive yet as independent to each other as possible to avoid collinearity. Thus, this study included the number of outpatients and emergency visits, number of home visits, the number of people received consulting or educating services, the number of children vaccinated, the number of public health issues reporting, and the number of physical examinations. Also based on the selection principle, the number of patients within management is better to be treated as an intermediate rather than output because it overlaps with some of the outputs. Similar to inputs, monetary output such as revenue was excluded. Outpatient and emergency visits were combined to one output as a common practice because the amount of emergency visits is very small and is not a major responsibility of PHCI [30]. Surgeries, discharged patients, and beds related outputs were not applicable in this study [43]. Table 1 lists all input and output variables with their definitions, as defined in the Statistical Survey System of Health Resources and Medical Services in Shenzhen (2020).

Inclusion criteria of PHCI with a specific year are located in Shenzhen city (including those newly established within the study period), submitted yearly report to the system. Data of PHCI with a specific year were excluded from the study if they meet one of the following criteria: number of health technicians was zero; numbers of all of the six output variables were zero. As a result, 37 PHCI-year data were excluded, leaving data of 3099 PHCI-years from 734 PHCI with at least one year of record ready to be analyzed in the next procedure.

The data uncertainty is negligible in this study. For each PHCI, the data for input and output variables were collected automatically through the system and were manually confirmed and signed by the manager in charge. Therefore, the measurement errors are low. The missing data due to certain variables not being reported were smaller than 2%, and the PHCI which had not been established in early years made the panel data to be unbalanced. For DEA and Tobit regression, this study evaluated the efficiencies for all PHCI whose data are available without missing variables. For the Malmquist index, this study calculated the indexes for all PHCI whose efficiencies for *t* and t−1 are both available. Therefore, the bias caused by missing data is negligible and there was no sampling variability since the data of all PHCI were used. In addition, since the number of DMUs in this study is as large as 734, the number is sufficiently large to generate an accurate evaluation of efficiencies and the Malmquist index.

For the Tobit regression model, data sources were the Shenzhen Health Statistical Information Report System and Shenzhen Health Statistics Yearbook (2016–2020). GPS coordinates of hospitals and PHCI in Shenzhen were acquired using Baidu Geocoding API.

Explanatory variables used for the Tobit regression were selected either based on previous studies that have been included to test the potential influence, or of special interest in this study [30,44,45]. Factors that influence the efficiency of PHCI were divided into two layers, the internal environmental factors and the external environmental factors [45]. Before fitting the data to the final model, a correlation analysis was performed to check collinearity between variables. In addition, the model was fitted through a stepwise manner from both directions using AIC of each model to remove variables that were neither contributing to the explanation of the data nor intriguing from the perspective of policy making. The independent variables included in the final model are listed in Table 1. MaxDEA 8.0 software (Beijing Realworld Software Company Ltd., Beijing, China) was used to obtain the result of DEA and Malmquist Index. Tobit regression and all other statistical analysis were performed on R 4.1.0 software (R Foundation for statistical computing, Vienna, Austria) [46]. R package censReg was used to conduct the Tobit regression [39].

## 3. Results

### 3.1. Descriptive Statistics of Inputs and Outputs

Table 2 and Table 3 list the descriptive statistics of inputs and outputs of the DEA model, respectively. From 2016 to 2020, the mean and the sum of all four inputs increased steadily. Among the outputs, the number of home visits maintained a steady increase, and the annual variations of the number of children vaccinated and the number of physical examinations were relatively small. However, the number of outpatient and emergency visits and the number of public health reporting decreased significantly in 2020. The number of people received consulting or educating services decreased year by year.

Figure 1 depicts the percentage of volume of inputs and outputs of PHCI in the whole medical and public health system consisting of PHCI and hospitals, demonstrating the portions of health resources and outputs taken up by PHCI in Shenzhen. The allocation of input resources within the medical and public health service system reflects the governmental strategy in combating the COVID-19 pandemic. The percentage of inputs of PHCI fluctuated within a small range from 2016 to 2019 without distinctive increase until a moderate increase happened in 2020. These input increases include the percentage of licensed (assistant) doctors, registered nurses, other health technicians, and equipment valued at more than RMB 10,000. Among them, the percentage of licensed (assistant) doctors has a significant escalation. Meanwhile, the proportion of outputs reflects the shouldered responsibility of demand as well as the impact of COVID-19 exerting to PHCI. Compared to the discernible increase of inputs in 2020, the increases of outputs of PHCI in 2020 are relatively small. A steady increase was observed in the percentage of home visits from 88.4% in 2019 to 94.6% in 2020. Meanwhile, the percentage of outpatients and emergency visits and the percentage of physical examinations remained constant throughout the observed time frame.

### 3.2. Results of DEA Model

Table 4 presents the results of TE, PTE and SE of PHCI in Shenzhen from 2016 to 2020. The mean of the TE, PTE, and SE of PHCI over the 5 years are 0.437, 0.542, and 0.809. This implies if the PHCI were running efficiently, an 45.8% of the inputs should have been decreased on average. Efficiency was the lowest in 2017 and the highest in 2018. Compared to other years, 2020 had a lower TE which is mainly due to the decrease in SE rather than PTE. Overall, the variation of PTEs across the five years is lower than that of SE. This could be interpreted as that larger PHCI are proportionally more vulnerable when the industry encounters a general recession, which is mainly reflected by a decrease in SEs.

### 3.3. Results of Malmquist Index

Table 5 and Table 6 present the Malmquist index results of annual geometric means and distribution of MI, EC, TC, PEC, and SEC of PHCI from 2017 to 2020. During the period from 2016 to 2019, the TFP and EC values of PHCI showed an upward trend, and TC showed a downward trend. Among them, TFP and TC are the maximum in 2019. However, from 2019 to 2020, the growth in TFP and the underlying TC values displayed a sharp decline, and the change of EC value was not obvious. Compared to TFP equal to 1.165 and TC equal to 1.304 in 2019, TFP in 2020 decreased considerably to 0.754, which means a 24.6% of productivity loss; the underlying TC in 2020 also decreased to 0.827, in comparison with the positive growth in TC (1.304) in 2019. In the epidemic year 2020, the values of EC and TC were both smaller than 1, among which deterioration in TC was larger. This indicates that the TFP in the epidemic year 2020 has a huge decline under the COVID-19 compared to the previous years. From a comprehensive perspective, the productivity loss in 2020 was attributable to both the decrease of EC and TC, and mainly due to the extensive degradation of TC. PEC and SEC seem to be highly correlated throughout the study period and they almost evenly decomposed EC. This indicates that the changes in pure technical efficiency and scale efficiency equally explain EC.

This result is in line with the frequency distribution of TFP from 2017 to 2020 (shown in Table 6). Over the period of 2017 to 2019, respectively, 48.2%, 38.8%, and 61.7% of PHCI in Shenzhen have TFP greater than one, indicating the growth in productivity; while compared to 2019, the TFP decreased sharply from 61.7% to 21.9% in 2020, which means only 21.9% of PHCI have TFP greater than one and 78.0% of the PHCI suffered from a regression of productivity.

### 3.4. Results of Tobit Regression

According to the preliminary results of stepwise analysis, registration type, revenue from financial subsidy, annual average increase of total asset, annual profit, ratio of nurses among health technicians were not significant contributors to the model and were thus removed from the final model. Ratio of nurses is also positively correlated with ratio of doctors, so the two variables were merge to one variable that is the ratio of doctors and nurses. The correlation analysis showed the risk of collinearity between the remaining variables is low. The descriptive statistics of variables as well as the Pearson correlation matrix for the final model of Tobit regression are included in Appendix A, Table A1.

Through the analysis of the Tobit model of the TE value (see Table 7), we found that the significant factors influencing the TE value of PHCI were revenue from operation, percentage of doctors and nurses in health technicians, ratio of doctors and nurses, Ln(service population), proportion of children in the service population, and numbers of PHCI within 1 kilometer. These factors are positively associated with the technical efficiency of PHCI. On the other hand, area, total asset, percentage of personnel salary in total expenditure, and average cost per outpatient or emergency visit were significant influencing factors of technical efficiency of PHCI, which are negatively associated with TE. The effects of established years, proportion of old people in the service population and per capita GDP, the effects of proportion of revenue from financial subsidy, and number of hospitals within 2 km were not statistically significant.

## 4. Discussion

The results of our analysis reflect considerable low levels of technical efficiency, pure technical efficiency, and scale efficiency of PHCI in Shenzhen, China, in 2017 and 2020. Compared to years before the epidemic, the productivity of PHCI reached the nadir during the COVID-19 pandemic along with the considerable technological efficiency change, despite the considerable inputs of health personnel and volume of health services. The significant factors that influence the growth of TE value of PHCI were the revenue from operation, percentage of doctors and nurses in health technicians, ratio of doctors and nurses, service population, proportion of children in the service population, and numbers of PHCI within 1 kilometer. These factors are positively associated with the technical efficiency of PHCI.

In 2017, the reduction of efficiency was mainly due to the severe drop of the number of people who received consulting or educating services and the number of physical examination. The unsatisfactory result in 2020 was mainly caused by the immense increase of inputs and the decline of almost all output variables, particularly the number of outpatients and emergency visits and the number of reporting, which was significantly impacted by the COVID-19 pandemic. We note that it is inappropriate to compare the efficiency score of DMUs between different efficiency studies, especially between models with different selections of inputs and outputs [47], given that this study is the only comprehensive study estimating and comparing the technical efficiency of PHCI in China before and during the epidemic. However, within this study, a separate analysis was performed with only the number of outpatients and emergency visits and the number of family visits as selected outputs. The even lower efficiency scores in this additional analysis indicate that including more output variables results in higher efficiency scores. The rationale behind this is simply because providing additional outputs allows PHCI to put more weight on outputs that the PHCI were favored of. Therefore, this study calls for enhancement of the evaluation system of PHCI in the post-pandemic era by including the additional responsibilities shouldered by PHCI, which mainly include the number of COVID-19 vaccinations, the number of people receiving epidemiological investigations, and the number of nucleic acid testing.

The study extended the DEA analysis by applying a Tobit regression model to compare the productivity of PHCI and to identify influencing factors that might lead to the existing inefficiency of PHCI over the time periods. It was found that the 24.6% decrease of productivity is due to the consequences of regression of technical efficiency change and technological efficiency change, though the decline in technological efficiency change is more severe. In addition, in 2019 and 2017, the higher MI were driven by the increase in technical efficiency change. These results match with previous results that technological efficiency change has a more profound influence on MI than technical efficiency change in PHCI in China [9,48]. This phenomenon could be explained by that the rank of productivity of a single PHCI is rather stable across years, but the shift of productivity frontier as a whole is more likely to happen in China where the productivity is highly associated with policy and administrative orders. For example, the shutdown and rejection of febrile patients in 2020 are the major causes of the technological efficiency change regression. While in the pre-pandemic era, a low technological efficiency change was traditionally perceived as the insufficient number of health technicians and under-utilization of medical equipment [30]. However, results in this study revealed that the increase in health personnel in 2020 failed to elevate technological efficiency change, which is in line with the disproportional increase in health personnel and volume of health services. This suggests that under-utilization is more likely to happen in the post-pandemic era, which calls for transformation of PHCI such as adopting tele-health technologies to maximize primary care delivery [49].

What is more, the results of Tobit regression provide insights in the effects of internal and external environmental factors. The negative effect of area and total asset on TE (p<0.001) confirmed that present medical resources are underutilized as suggested in previous studies [30]. The effect of revenue from operation and percentage of personnel salary in total expenditure indicate that a more robust financial structure with a good flow of operating income is associated with a higher TE. Average cost per visit is negatively associated with TE which was congruent with previous analysis [44]. Similarly, higher percentage of doctors and nurses and higher ratio of doctors to nurses are positively associated with TE [50]. In addition, the service population size is positively associated with TE, as well as the percentage of children and old people since both groups are important targets of services [30]. The effect of city level is not available in this setting. However, as a proxy of developmental status of regions, per capita GDP did not have any significant effects on TE, which indicates few variations of TE geographically. Availability of other medical institutes was assumed to decrease TE as previously suggested [44], which is also the case for hospitals in this study. Despite of this, the presence of nearby PHCI has a positive effect on TE. The mechanism that might explain this phenomenon is that the clustering of PHCI is a representation of the elevated knowledge and preference of search for medical services from PHCI.

In May 2021, a new administrative order in Shenzhen was implemented that PHCI shall reject patients with the 10 symptoms of COVID-19 (fever, dry cough, fatigue, smell loss, nasal congestion, runny nose, sore throat, conjunctivitis, myalgia, and diarrhea) [51]. PHCI that violated the administrative order will receive punishment in the form of business suspension. Affected by this, it is expected that in 2021, there will be a further decline in the volume of service outputs. China is not the only country that is facing the issue. Ensuring operational continuity and practice resilience in PHCI becomes a challenge worldwide [52]. Meanwhile, as the order of increasing the number of physicians and general practitioners persists, production inputs of PHCI will maintain its increasing tendency. Therefore, a sharp decrease in efficiency of the regular services delivered by PHCI could be anticipated. Furthermore, since the financial subsidy allocated to PHCI is determined by the volume of services, the business situation would be pessimistic due to the reduction of number of visits and elevated personnel costs.

Based on the aforementioned analysis, we believe that there are three key aspects to improve the technical efficiency of PHCI in both routine healthcare and infectious disease control, to better prepare for a future pandemic. First, the newly emerged workloads of COVID-19 vaccinations, epidemiological investigations, and nucleic acid testing need to be incorporated into the evaluation framework of PHCI. Supportive financial allocation needs to be matched with and be proportional to the volume of the workloads to ensure operational continuity. Second, persistent efforts in the training of healthcare suppliers are needed as the foundation to achieve the integrated health delivery system. Optimization of expertise distribution with a supporting personnel salary system are encouraged. Third, increasing the number of health technicians is crucial to realize universal health coverage, but it is also necessary to promote the adoption of new forms of health care delivery such as tele-health to fully mobilize expertise resources.

This study has several significant strengths. First, to our knowledge, this study is the first to perform a comprehensive analysis of the service efficiency change of PHCI and the underlying influencing factors before and during the pandemic using DEA, MI, and Tobit model in China. This research includes the originality of the data and its time span that allows a comparison between the first year of the COVID outburst and the pre-COVID era. Second, the comprehensive and thorough selection of health resources input and output variables fully demonstrated the mechanism of DEA model and comprehensively reviewed the responsibilities of PHCI under the up-to-date medical system in China. Third, this study thoroughly matched model outputs with real-world policy changes and identified clear causes of the efficiency changes that are in line with existing research. Fourth, the Tobit regression in this study covers a variety of internal and external environmental factors, which gives insights to internal management and external control by the government. Fifth, this study provided novel and practical suggestions for improvement of PHCI based on real-world evidence that can serve as a reference for Shenzhen city as well as other cities or regions worldwide.

Nevertheless, the study is subject to some limitations. First, the scope of this research is within the Shenzhen city, a relatively developed city, which made the results conductive to cities with middle-to-high incomes, but less informative to underdeveloped areas with fewer health care resources. Second, in the Tobit regression model, more internal variables such as the characteristics of the leadership and waiting time for service could be introduced, as suggested previously [53,54]. Data availability restricted us from further exploring these factors. Another limitation of this study is not considering the integrality of variables. Adjustments to the classic DEA method have been made to account for the fact that some input and output variables may only take integer values [55,56]. It could result in a more realistic evaluation of DMU efficiencies, especially when the number of DMUs and the magnitude of variables are small. However, this issue is considered as negligible due to the large number of DMUs and the large values of output variables in this study that mitigate the problem. In addition, the difference of the efficiencies estimated by the two models is small [56], which guaranteed our results to be capable of reflecting the trends of the overall performance of PHCI. Nevertheless, the results from this study are authentic and representative to yield meaningful insights for improvements of PHCI.

In conclusion, our study shows the technical efficiency, pure technical efficiency, scale efficiency, productivity changes, and influencing factors of PHCI in Shenzhen, China, throughout three phases of the operation, regular business, shutdown due to COVID-19, and a gradual recovery of business. The technical efficiency significantly declines along with the COVID-19 outbreak in Shenzhen, China. In total, the productivity of PHCI decreased by 24.6% in 2020 with the deterioration of underlying technical efficiency change and technological efficiency change, regardless of the immense inputs of health resources. Transformation of PHCI such as adopting tele-health technologies to maximize primary care delivery is needed to optimize utilization of health resource inputs. Future research is recommended to examine the performance of PHCI on a larger scale to evaluate the progress of the transformations of PHCI in the post-pandemic era. The study sheds light on improving the performances of PHCI in China to respond to the current epidemiologic transition and future epidemic outbreaks more effectively, and to promote the national strategy of Healthy China 2030.

## 5. Conclusions

The findings of this study suggests that there has been a significant decline in technical efficiency during the COVID-19 outbreak in Shenzhen, China. The decline is evident in both underlying technical efficiency change and technological efficiency change, despite the considerable inputs in health resources. To optimize utilization of health resource inputs, transformation of PHCI such as adopting tele-health technologies to maximize primary care delivery is needed. This study brings valuable insights to improve the performances of PHCI in China in response to the current epidemiologic transition and future epidemic outbreaks more effectively, and to promote the national strategy of Healthy China 2030.

## Figures and Tables

**Figure 1 ijerph-20-04453-f001:**
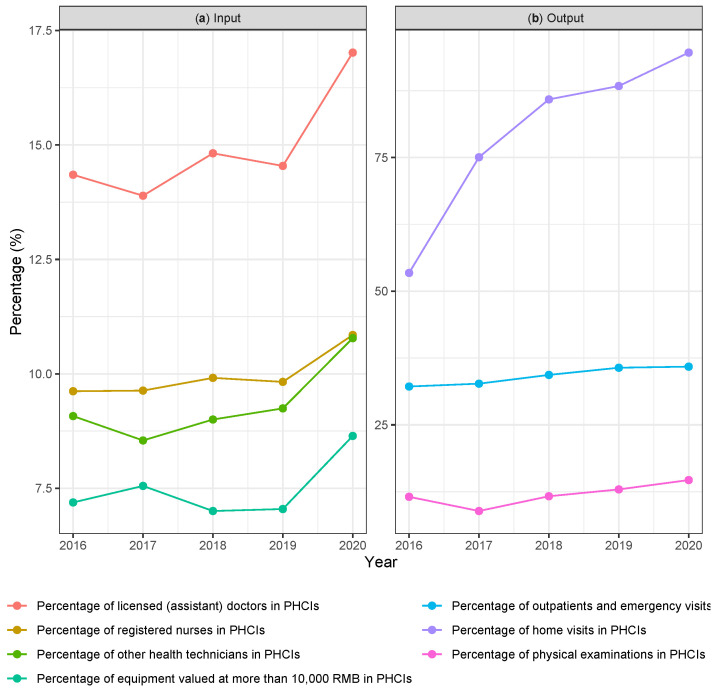
The percentage of volume of comparable input and output variables contributed by PHCI in Shenzhen from 2016 to 2020. (**a**) Percentages of input variables. (**b**) Percentages of output variables.

**Table 1 ijerph-20-04453-t001:** Definitions of input, output, and influencing factors.

Category	Variable	Definition	Unit
Input	Number of licensed (assistant) doctors	The number of people who have obtained the practicing certificates of doctors and are actually engaged in clinical or supervision work.	person
	Number of registered nurses	The number of people who have obtained the practicing certificates of nurses and are actually engaged in clinical or supervision work.	person
	Number of other health technicians	Including health professionals such as pharmacists, inspection technicians, imaging technicians, health supervisors, and intern doctors.	person
	Number of equipment valued at more than RMB 10,000	The number of all equipment (with their auxiliaries) above RMB 10,000, including medical equipment and logistics equipment.	set
Output	Number of outpatients	The number of people with outpatient registration and have received actual diagnosis and treatment.	person
	Number of home visits	The number of people with a family hospital bed and received door-to-door health services such as diagnosis, treatment and nursing.	person
	Number of people received consulting or educating services	The number of people received health education or benefited from public health consultation activities during the year	person
	Number of children vaccinated	The number of children vaccinated according to the National Immunization Program for children aged 0–6 years.	person
	Number of public health issues reporting	The number of reports of cases of infectious diseases and public health emergencies in the year as well as number of reports of health supervision and co-management in the year.	case
	Number of physical examinations	The number of general physical examinations and single-item physical examination.	person
Independent	Established years	The number of years from the establishment to the year of reporting.	year
	Area	The area of the PHCI, including building area and rental area that are occupied for operation but excluding residential area for personnel.	100 m${}^{2}$
	Total asset	The original value of total asset that the PHCI possesses at the end of the fiscal year.	RMB 1,000,000
	Revenue from operation	Including medical service revenue, medicine revenue and public health revenue. Also equals to total revenue minus revenue from financial subsidy.	RMB 1,000,000
	Proportion of revenue from financial subsidy	The proportion of revenue from financial subsidy in the total revenue.	-
	Proportion of personnel salary in total expenditure	The proportion of personnel basic salary, performance-related salary, allowance, social insurance in total expenditure.	-
	Proportion of doctors and nurses in health technicians	The proportion of licensed doctors, assistant doctors and nurses in all health technicians.	-
	Ratio of doctors and nurses	The ratio of the number of licensed doctors and assistant doctors over the number of nurses. (Ratios that were infinite were coerced to the maximum ratio 11.)	-
	Average cost per outpatient or emergency visit	The revenue from outpatient and emergency visits divided by the number of outpatient and emergency visits.	RMB 10
	Service population	The number of permanent residents that are covered by the service of the PHCI at the end of the year.	person
	Proportion of children in the service population	The proportion of children aged 0–6 in the service population.	-
	Proportion of old people in the service population	The proportion of people aged above 65 years old in the service population.	-
	Number of PHCI within 1 km	The number of PHCI that locate within 1 km to the PHCI in straight line distance.	number
	Number of hospitals within 2 km	The number of hospitals that locate within 2 km to the PHCI in straight line distance.	number
	Per capita GDP	The annual per capita GDP of the district where the PHCI locate.	RMB

**Table 2 ijerph-20-04453-t002:** Descriptive statistics of inputs of the DEA model.

Year	Measure	Number of Licensed (Assistant) Doctors	Number of Registered Nurses	Number of Other Health Technicians	Number of Equipment Valued at More Than RMB 10,000
2016	Median	5	4	1	9
	Mean	5.67	4.77	1.75	11.44
	SD	3.86	3.10	2.80	11.14
	Sum	3308	2780	1023	6668
2017	Median	5	4.5	1	10
	Mean	5.95	4.88	1.77	13.42
	SD	4.00	3.10	2.64	13.10
	Sum	3570	2929	1060	8050
2018	Median	6	5	1	11
	Mean	6.89	5.46	2.02	14.19
	SD	4.59	3.41	3.15	14.12
	Sum	4224	3348	1237	8700
2019	Median	6	5	1	12
	Mean	7.04	5.55	2.09	15.34
	SD	4.86	3.45	2.89	15.47
	Sum	4556	3592	1349	9928
2020	Median	7	5	1	14
	Mean	7.89	5.68	2.10	18.22
	SD	5.26	3.38	2.67	17.17
	Sum	5256	3786	1397	12,135

**Table 3 ijerph-20-04453-t003:** Descriptive statistics of outputs of the DEA model.

Year	Measure	Number of Outpatients and Emergency Visits	Number of Home Visits	Number of People Received Consulting or Educating Services	Number of Children Vaccinated	Number of Physical Examinations	Number of Public Health Issues Reporting and Surveillance
2016	Median	38,975	0	6484	7668	75	6
	Mean	45,330.50	1905.96	27,468.42	8898.34	1442.47	35.81
	SD	35,793.93	5649.98	43,495.73	8367.27	4361.42	122.86
	Sum	26,427,684	1,111,177	16,014,086	5,187,733	840,959	20,875
2017	Median	39,541.5	30	2851	9320.5	0	18
	Mean	46,150.03	3434.13	15,925.67	11,182.82	1050.27	114.67
	SD	34,891.95	8165.93	33,282.11	12,025.18	3236.32	289.39
	Sum	27,690,017	2,060,476	9,555,404	6,709,692	630,161	68,804
2018	Median	41,934	68	1696	9216	21	31
	Mean	48,169.27	3773.03	10,919.86	10,600.84	1559.09	156.24
	SD	35,769.13	8353.40	28,233.14	11,936.76	7232.63	382.85
	Sum	29,527,760	2,312,869	6,693,874	6,498,316	955,724	95,773
2019	Median	46,489	45	1538	8627	190	70
	Mean	52,144.95	4006.52	7499.91	10,009.79	1784.23	211.11
	SD	38,638.56	8817.18	21,154.83	10,976.57	5789.74	349.63
	Sum	33,737,784	2,592,217	4,852,442	6,476,333	1,154,399	136,585
2020	Median	34,537.5	55	1314	8277	413	43
	Mean	39,796.34	4176.89	6889.90	9534.68	1602.97	102.65
	SD	28,303.55	8994.06	20,058.79	9447.83	3155.56	163.49
	Sum	26,504,364	2,781,812	4,588,676	6,350,100	1,067,575	68,367

**Table 4 ijerph-20-04453-t004:** The efficiency of Primary Health Care Institutions (PHCI) in Shenzhen, China, from 2016 to 2020.

Year	TE ${}^{1}$		PTE ${}^{2}$		SE ${}^{3}$	
	Mean	Number of Efficient PHCI ${}^{4}$	Mean	Number of Efficient PHCI ${}^{5}$	Mean	Number of Efficient PHCI ${}^{6}$
2016	0.470	45 (7.72%)	0.554	85 (14.58%)	0.845	45 (7.72%)
2017	0.328	28 (4.67%)	0.466	71 (11.83%)	0.726	28 (4.67%)
2018	0.545	48 (7.83%)	0.599	90 (14.68%)	0.903	48 (7.83%)
2019	0.452	35 (5.41%)	0.547	86 (13.29%)	0.829	35 (5.41%)
2020	0.392	37 (5.56%)	0.541	104 (15.62%)	0.746	40 (6.01%)

^1^ TE: Technical efficiency; ^2^ PTE: Pure technical efficiency; ^3^ SE: Scale efficiency; ^4^ “efficient” means TE is equal to 1; ^5^ “efficient” means PTE is equal to 1; ^6^ “efficient” means SE is equal to 1.

**Table 5 ijerph-20-04453-t005:** Annual geometric means of the Malmquist index of PHCI in Shenzhen from 2017 to 2020.

Year(t)	TFP(t−1,t) ${}^{1}$	EC(t−1,t) ${}^{2}$	TC(t−1,t) ${}^{3}$	PEC(t−1,t) ${}^{4}$	SEC(t−1,t) ${}^{5}$
2017	1.040	0.655	1.587	0.776	0.845
2018	0.913	2.065	0.442	1.466	1.409
2019	1.165	0.893	1.304	0.943	0.946
2020	0.754	0.913	0.827	1.000	0.913
2017-2020	0.968	1.131	1.040	1.046	1.028

^1^ TFP: Total factor productivity change; ^2^ EC: Technical efficiency change; ^3^ TC: Technological efficiency change; ^4^ PEC: Pure technical efficiency change; ^5^ SEC: Scale efficiency change.

**Table 6 ijerph-20-04453-t006:** Frequency distribution of the Malmquist index of PHCI in Shenzhen from 2017 to 2020.

Year	Range	TFP(t−1,t) ${}^{1}$	EC(t−1,t) ${}^{2}$	TC(t−1,t) ${}^{3}$	PEC(t−1,t) ${}^{4}$	SEC(t−1,t) ${}^{5}$
2017	>1	273 (48.2%)	128 (22.6%)	512 (90.3%)	163 (28.8%)	165 (29.1%)
	=1	0 (0.0%)	14 (2.5%)	0 (0.0%)	35 (6.2%)	14 (2.5%)
	<1	294 (51.9%)	425 (75.0%)	55 (9.7%)	369 (65.1%)	388 (68.4%)
2018	>1	223 (38.8%)	492 (85.6%)	14 (2.4%)	413 (71.8%)	454 (79.0%)
	=1	0 (0.0%)	10 (1.7%)	0 (0.0%)	39 (6.8%)	10 (1.7%)
	<1	352 (61.2%)	73 (12.7%)	561 (97.6%)	123 (21.4%)	111 (19.3%)
2019	>1	364 (61.7%)	179 (30.3%)	561 (95.1%)	214 (36.3%)	173 (29.3%)
	=1	0 (0.0%)	16 (2.7%)	0 (0.0%)	47 (8.0%)	16 (2.7%)
	<1	226 (38.3%)	395 (67.0%)	29 (4.9%)	329 (55.8%)	401 (68.0%)
2020	>1	137 (21.9%)	211 (33.6%)	87 (13.9%)	276 (43.9%)	199 (31.6%)
	=1	1 (0.2%)	10 (1.6%)	1 (0.2%)	38 (6.0%)	10 (1.6%)
	<1	489 (78.0%)	408 (64.9%)	539 (86.0%)	315 (50.1%)	420 (66.8%)

^1^ TFP: Total factor productivity change; ^2^ EC: Technical efficiency change; ^3^ TC: Technological efficiency change; ^4^ PEC: Pure technical efficiency change; ^5^ SEC: Scale efficiency change.

**Table 7 ijerph-20-04453-t007:** Result of Tobit regression (N = 2944).

Variable	Coefficient	SE ${}^{1}$	*p*-Value ${}^{2}$	95% CI ${}^{3}$
Intercept	−0.105	0.199	0.597	(−0.496, 0.285)
Established years	0.001	0.001	0.201	(−0.001, 0.003)
Area	−0.005	0.001	<0.001 ***	(−0.006, −0.003)
Total asset	−0.002	0.000	<0.001 ***	(−0.003, −0.001)
Revenue from operation	0.014	0.002	<0.001 ***	(0.010, 0.017)
Percentage of revenue from financial subsidy	−0.023	0.024	0.339	(−0.070, 0.024)
Percentage of personnel salary in total expenditure	−0.163	0.025	<0.001 ***	(−0.213, −0.114)
Percentage of doctors and nurses in health technicians	0.202	0.050	<0.001 ***	(0.104, 0.299)
Ratio of doctors and nurses	0.033	0.003	<0.001 ***	(0.028, 0.039)
Average cost per outpatient or emergency visit	−0.005	0.001	<0.001 ***	(−0.006, −0.003)
Ln(Service population)	0.036	0.007	<0.001 ***	(0.023, 0.049)
Percentage of children in the service population	0.264	0.052	<0.001 ***	(0.161, 0.367)
Percentage of old people in the service population	0.123	0.088	0.165	(−0.051, 0.296)
Number of PHCI within 1 km	0.004	0.002	0.039 *	(0.000, 0.008)
Number of hospitals within 2 km	−0.002	0.002	0.207	(−0.005, 0.001)
Ln(Per capita GDP)	0.001	0.015	0.926	(−0.027, 0.030)
Ln(SigmaMu)	−2.035	0.044	<0.001 ***	(−2.122, −1.949)
Ln(SigmaNu)	−1.558	0.014	<0.001 ***	(−1.585, −1.530)

^1^ SE: Scale efficiency; ^2^ *p*-Value: *** *p* < 0.001, * *p* < 0.05; ^3^ 95% CI: 95% Confidence interval.

## Data Availability

Restrictions apply to the availability of these data. Data was obtained from Shenzhen Health Statistical Information Report System and Shenzhen Health Statistics Yearbook (2016-2020), and are not available to public.

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
