# Peer review of "Technical Efficiency Evaluation of Primary Health Care Institutions in Shenzhen, China, and Its Policy Implications under the COVID-19 Pandemic"

_ijerph, 2023, doi:10.3390/ijerph20054453_

Round 1

Reviewer 1 Report

- In the variables list of current study, most variables such as Number of licensed doctors (X1), Number of registered nurses (X2), Number of other health technicians (X3), Number of equipment valued at more than 10,000 RMB (X4), Number of outpatients (Y1), Number of home visits (Y2), Number of people received consulting or educating services (Y3), Number of children vaccinated (Y4), Number of public health issues reporting (Y5), Number of physical examinations (Y6), are integer. Does the proposed approach have the ability to be used in the presence of integer values? For more details see the following references:

Kuosmanen, T., & Matin, R. K. (2009). Theory of integer-valued data envelopment analysis. European Journal of Operational Research, 192(2), 658-667.

Lozano, S., & Villa, G. (2006). Data envelopment analysis of integer-valued inputs and outputs. Computers & Operations Research, 33(10), 3004-3014.

- The authors should clearly explain and justify the reason for using Classic DEA approach instead of Network DEA and Dynamic DEA approaches.

- Advantages and benefits of the proposed approach should be given in detail. Also, the research gaps and the novelty of this study is not clear.

- The characteristics of current research should be highlighted in the comparative table of literature review from both aspects of theoretical and application.

- Generally, real data are tainted by uncertainty. The authors should discuss the proposed approach under data uncertainty.

- The authors should discuss on the limitations of the study. Also, the authors should discuss on the generalization of the results of the study.

Reviewer 2 Report

This manuscript analyses technical efficiency of primary health care in Shenzhen, China. The topic is of particular interest for the journal IJERPH. The main original points of the research include the originality of the data and their time span that, although limited to only 5 years, allows a comparison between the first year of COVID outburst and the pre-COVID era. In overall, the manuscript is very well written: I have just few minor issues listed below.
Lines 172 and 211: reference link is missing.
Authors often write 'Malmquist index model'. This is not correct, since they are index numbers, not a model. Therefore, I would eliminate the word 'model' each time reference is made to Malmquist index numbers.
Line 209: referring to vector beta, instead of the term 'effects', it seems more appropriate to me to use the term 'coefficients'.
Line 302: it would be useful for the reader ro recall the meaning of the acronyms TE, PTE and SE. Similarly, at line 313 for the acronyms Mi, EC, TC, PEC and SEC.
Table 4: caption is missing. In the caption, I would add what is meant with 'efficient'. I suppose that it is meant that technical efficiency is equal to 1.
Table 5: the line label 'Mean' could be replaced with '2017-2020' for increasing clarity.
Table 7: acronyms SE and CI should be defined in the caption. Also, parameters Ln(SigmaMu) and Ln(SigmaNu) are undefined. Please state what they represent in reference to the model formulation reported in Section 2.3.

Round 2

Reviewer 1 Report

Accept in present form